# Immunological Patient Stratification in Myalgic Encephalomyelitis/Chronic Fatigue Syndrome

**DOI:** 10.3390/jcm13010275

**Published:** 2024-01-03

**Authors:** Johanna Rohrhofer, Lisa Hauser, Lisa Lettenmaier, Lena Lutz, Larissa Koidl, Salvatore Alessio Gentile, Davide Ret, Michael Stingl, Eva Untersmayr

**Affiliations:** 1Institute of Pathophysiology and Allergy Research, Center for Pathophysiology, Infectiology and Immunology, Medical University of Vienna, 1090 Vienna, Austria; johanna.rohrhofer@meduniwien.ac.at (J.R.); larissa.koidl@meduniwien.ac.at (L.K.); salvatore.gentile@meduniwien.ac.at (S.A.G.); davide.ret@tuwien.ac.at (D.R.); 2Division of Macromolecular Chemistry, Institute of Applied Synthetic Chemistry, Vienna University of Technology, 1060 Vienna, Austria; 3Facharztzentrum Votivpark, 1090 Vienna, Austria; ordination@neurostingl.at

**Keywords:** myalgic encephalomyelitis/chronic fatigue syndrome, immunology, immunodeficiencies, mucosal barrier function, patient stratification, post-viral fatigue, intestinal barrier leakage, debilitating disease

## Abstract

Myalgic encephalomyelitis/chronic fatigue syndrome (ME/CFS) is a complex disease characterized by profound fatigue, post-exertional malaise (PEM), and neurocognitive dysfunction. Immune dysregulation and gastrointestinal symptoms are commonly observed in ME/CFS patients. Despite affecting approximately 0.89% of the general population, the underlying pathophysiological mechanisms remain poorly understood. This study aimed to elucidate the relationship between immunological characteristics and intestinal barrier function in ME/CFS patients. ME/CFS patients were stratified into two groups based on their immune competence. After documentation of detailed medical records, serum and plasma samples were collected for the assessment of inflammatory immune mediators and biomarkers for intestinal barrier integrity by ELISA. We found reduced complement protein C4a levels in immunodeficient ME/CFS patients suggesting a subgroup-specific innate immune dysregulation. ME/CFS patients without immunodeficiencies exhibit a mucosal barrier leakage, as indicated by elevated levels of Lipopolysaccharide-binding protein (LBP). Stratifying ME/CFS patients based on immune competence enabled the distinction of two subgroups with different pathophysiological patterns. The study highlights the importance of emphasizing precise patient stratification in ME/CFS, particularly in the context of defining suitable treatment strategies. Given the substantial health and socioeconomic burden associated with ME/CFS, urgent attention and research efforts are needed to define causative treatment approaches.

## 1. Introduction

Myalgic encephalomyelitis/chronic fatigue syndrome (ME/CFS) is a multifactorial disease associated with a profound and disabling fatigue lasting for longer than six months. ME/CFS is accompanied by a systemic exercise intolerance and worsening of symptoms after physical or mental exertion, which is termed post-exertional malaise (PEM), as well as neurological and/or cognitive dysfunction, orthostatic dysregulation, and various other conditions [1]. Besides immune dysregulation, autoimmunity or immunodeficiencies (IDs) are also frequently observed in ME/CFS patients [2]. Based on data from public databases, 0.89% of the general population are estimated to suffer from ME/CFS, with an approximately 1.5- to 2-fold higher prevalence in women [3]. Diverse mechanisms of disease development are discussed ranging from metabolic and endocrinological disorders to immune-related mechanisms. While a subset of patients develops ME/CFS over a long time span without being able to name a certain disease-initiating event, a large subgroup of patients develops post-infectious ME/CFS, mentioning especially viral infections as the main trigger [4]. Hypotheses on disease triggers range from asymptomatic viral infections, explaining the subgroup of patients who are not able to identify the disease initiating event, to a genetically greater host susceptibility to infections, where latent infected cells are associated with immune dysfunction and post-viral fatigue [5]. However, a definite disease-initiating pathway, which all ME/CFS patients have in common, has not been described yet, leaving the definition of pathomechanisms challenging.

With the outbreak of the COVID-19 pandemic, we are in a unique situation to monitor and investigate the mechanisms of early post-viral fatigue, which might ultimately lead to ME/CFS. SARS-CoV-2 infection, independent of the severity of the acute phase, results in 1–10% of cases in sequelae termed Long-COVID (LC) or post-acute sequelae of COVID-19 (PASC) [6]. While some LC symptoms are associated with organ damage due to viral host invasion [7], a sizeable subgroup of patients develops post-viral fatigue. Alarmingly, first estimations predict a doubling of ME/CFS patients due to SARS-CoV-2 within the next years [8], making an increase in ME/CFS research effort a matter of uttermost importance. Translating LC research into ME/CFS research will be crucial to improve patient health care and improve the definition of disease onset and pathogenesis in ME/CFS.

Despite huge knowledge gaps in disease onset and fatigue development, similarities in ME/CFS and LC are documented. Both ME/CFS and LC are often associated with altered immune functions and a dysregulated mucosal barrier. Tumor necrosis factor (TNF)-alpha, Interleukin (IL)-6, and IL-8 were previously shown to be related to fatigue severity in ME/CFS [9]. However, the results are often inconsistent throughout the literature depending on the study design and subsequent ME/CFS patient stratification. A higher prevalence of Mannose-binding lectin (MBL) deficiency is found among LC patients with persistent severe fatigue and PEM [10]. This is also in line with ME/CFS patients developing the disease after EBV infection [2,11]. It is noteworthy that MBL is essential for the innate immune response, which acts as a first line defense to prevent viral invasion or replication. It has been suggested to regulate NK cell function [12] and works as a pattern recognition receptor [13] by binding and opsonizing viral proteins, i.e., the S and N proteins of SARS-CoV-2 [14,15]. Furthermore, MBL deficiency is associated with an increased susceptibility to infection and genetically determined low levels of MBL can predispose people to viral diseases, such as COVID-19 [13]. Also, LC patients showed altered humoral responses to distinct herpesviruses including EBV. In accordance with these data, our group recently demonstrated an association between EBV reactivation and LC fatigue, with the presence of EBV DNA in 50% of LC fatigue patients and only 20% in fully convalescent COVID-19 patients [16]. These mechanisms support the hypothesis that in a subset of ME/CSF and post-infectious syndrome patients, an inadequate immune activation during a viral acute phase may contribute to the disease’s development.

When focusing on an inadequate immune response during an acute infection, it is important to elucidate the role of mucosal barrier function. As mucosae protect the human body from the outside environment, failed protection has been suggested to be involved in the disease mechanisms of post-infectious syndromes. This hypothesis is based on frequently observed gastrointestinal complaints in ME/CFS patients. While recurring infections can be associated with inadequate host defense, most gastrointestinal-affected patients suffer from symptoms similar to irritable bowel syndrome (IBS) or gastrointestinal mast cell activation disorders [1,17], without being diagnosed with inflammatory bowel disease (IBD) or other detectable inflammatory gastrointestinal disorders. The gut microbiome gained much attention in ME/CFS research with proposed pathomechanisms regarding gut dysbiosis and subsequent altered gut–brain axis activity but also increased gut permeability with concomitant bacterial translocation enabling a chronic, low-grade gastrointestinal inflammation. SARS-CoV-2 has been demonstrated to affect gastrointestinal barrier function via ACE-2 receptor binding, and viral particles are found in patients with gastrointestinal inflammation [18], providing a possible mechanism for intestinally triggered post-viral fatigue. Also, other discussed viral triggers of ME/CFS, like influenza virus A, coxsackievirus, and several members of the Herpesviridae family are known to break mucosal tissue barriers and impair epithelial and microbial barrier functions [19,20].

Although ME/CFS was described as a disease by the WHO more than 50 years ago and ME/CFS prevalence is increasing due to SARS-CoV-2, the exact causes of the disease remain elusive. The lack of well-characterized patient stratification not only hinders the identification of biomarkers, which are urgently needed in diagnostics, but also ultimately impairs the definition of causative treatment approaches. Until today, there is no cure or approved treatment strategy for ME/CFS [21].

In this study, we aim to evaluate immunological and mucosal barrier-related insights in an Austrian ME/CFS cohort. Understanding immune dysfunction and the impact of related factors, like immunodeficiency and mucosal barrier disruption, can lead to targeted interventions and therapies to alleviate symptoms and improve patient outcomes. We offer insights into ME/CFS subgroups, with the ultimate goal of improving the medical care situation of ME/CFS and post-infectious fatigue patients.

## 2. Materials and Methods

### 2.1. Study Cohort and Sampling Procedure

All participants provided written informed consent before study inclusion and were recruited between June 2021 and November 2022, including a break from February 2022 to July 2022 due to COVID-19-related access restrictions to the laboratory facility. As shown in Figure 1, 39 ME/CFS patients and 19 healthy sex- and age-matched control participants were recruited. ME/CFS was diagnosed by a specialized neurologist based on the Institute of Medicine (IOM) criteria [22] after exclusion of other medical conditions associated with profound fatigue. The IOM criteria define that the following three symptoms and at least one of two additional manifestations are required for diagnosis. The three required symptoms are the following: (1) a substantial reduction/impairment in the ability to engage in pre-illness levels of activity, (2) PEM, and (3) unrefreshing sleep. Additional manifestations include cognitive impairment and/or orthostatic intolerance. All included ME/CFS patients suffered from an EBV-related onset of the disease. ME/CFS patients were further divided in a group of patients without immunodeficiencies (ME/CFS − ID, n = 19) and patients with immunodeficiencies (ME/CFS + ID, n = 20). The immunological evaluation of the ME/CFS patients (Figure 1) was conducted during diagnostic evaluations prior to study participation as part of the clinical routine workup according to available definitions for immunodeficiencies (European Society for Immunodeficiencies, ESID). A differential blood count, leukocyte subtyping, and quantification of immunoglobulins and IgG subclasses, as well as a complement analysis, was performed, as discussed elsewhere [2]. Patients were tested for C3 and C4 values. However, only C3 deficiencies were diagnosed. Additional to defined laboratory values, an indicative medical history of frequent recurrent and/or severe infections was required for the diagnosis of immunodeficiencies. ME/CFS patients without immunodeficiencies were within the reference of the norm values. ME/CFS patients with immunodeficiencies were on average 41.2 years old (±12.6 years) and 75% female, while ME/CFS patients without immunodeficiencies were on average 38.4 years old (±10.8 years) and 84.2% female. Healthy control participants were on average 43.1 years old (±13.0 years) and 73.7% female (Table 1). Besides one severely affected patient in the ME/CFS + ID group, all ME/CFS patients were mildly to moderately affected by the disease. The definition of the severity degrees is based on international consensus criteria [23] and they are defined as a significant reduction in a patients’ activity level. They can be divided into “mild” (~50% reduction in pre-illness activity level), “moderate” (mostly housebound), “severe” (mostly bedridden), or “very severe” (totally bedridden and need help with basic functions). Healthy controls were excluded when suffering from neurological, immunological, or psychiatric pre-existing conditions. In total, 45% of ME/CFS patients with immunodeficiencies, 42.1% of ME/CFS patients without immunodeficiencies, and 84.2% of healthy controls had COVID-19 prior to study participation. All participants, including healthy controls, were at least 12 weeks past an acute SARS-CoV-2 infection. Before the sampling of biological material, participants were asked to give a detailed medical record on background and individual symptoms (Table 2). Blood was collected in serum and plasma/EDTA vials on the day of study participation at the Institute for Pathophysiology and Allergy Research, Medical University of Vienna. After centrifugation with 2000× *g* for 10 min, serum and plasma were retrieved and stored at −20 °C until further analysis.

### 2.2. Measurement of Disease-Related Parameters

As an important aspect of the study was to improve ME/CFS diagnosis and patient health care, we focused on parameters feasible in routine diagnostics. All parameters were obtained from plasma or serum samples and measured by commercially available enzyme-linked immunosorbent assays (ELISAs). Markers of interest were chosen to determine (1) chronic, low-grade inflammation and immune dysfunction, (2) mucosal barrier disruption, and (3) mast cell-related disorders. To determine chronic low-grade inflammation and immune dysfunction, the pro-inflammatory IL-1-beta, IL-6, TNF-alpha, IL-8, IL-33, and IFN-gamma were measured, as well as complement protein C4a and fibroblast growth factor 21 (FGF-21). The evaluation of mucosal barrier disruption was performed by measuring lipopolysaccharide-binding protein (LBP) and the soluble cluster of differentiation 14 (sCD14), as well as fatty acid binding protein-2 (intestinal, I-FABP) and IgG endotoxin Core antibodies (Endotoxin Core IgG). Levels of eosinophil cationic protein (ECP) and eosinophil-derived neurotoxin (EDN) have been determined to evaluate eosinophil activation. All markers and respective ELISA kits are listed in Appendix A (Table A1). ELISAs were performed according to the respective manufacturer’s protocols. If requested by the protocol, assays were validated prior to the main analysis. All samples were measured in duplicate and compared to internal standards. More detailed information on the performance of each assay can be found in the respective protocols (Table A1). All reagents used were provided by the respective ELISA kits or prepared according to the manufacturer’s instructions. To briefly summarize the procedure, 96-well microplates were coated with an antigen-specific capture antibody. After incubation and washing, serum or plasma samples were applied. All used detection antibodies were biotinylated and detected by a streptavidin/horse radish peroxidase conjugate. Tetramethylbenzidine and an acidic stopping solution were used for the colorimetric reaction. Absorbance was measured at 450 nm using an ELISA Reader Infinite m200 PRO (Tecan, Männedorf, Switzerland). A four-parameter logistic (4PL) curve was used to analyze antibody concentrations after subtracting levels detected in blank wells as background values. No time-to-assay effects were detected, which might have been associated with the COVID-19-related recruiting break of our study.

### 2.3. Statistical Analyses

Quantitative variables are reported as mean ± standard deviation (SD) if not stated otherwise. All acquired data sets were tested for normal distribution by using the Kolmogorov–Smirnov test. If normally distributed, data sets were compared by one-way ANOVA. The confidence level was adjusted by Tukey’s multiple comparison test and variances were computed for each comparison. If normal distribution was not given, data sets were compared by the Kruskal–Wallis test. The confidence level was adjusted by Dunn’s multiple comparisons of all columns and individual variances were computed for each comparison. Multiplicity adjusted *p*-values < 0.05 were considered statistically significant. Data analysis was conducted by GraphPad Prism 9 software (GraphPad, San Diego, CA, USA).

## 3. Results

### 3.1. Evaluation of Pro-Inflammatory Immune Mediators

To detect signs of chronic low-grade inflammation and evaluate respective markers in ME/CFS patients, pro-inflammatory and mucosal barrier integrity-related cytokines IFN-gamma, TNF-alpha, IL-6, IL-1-beta, IL-8, and IL-33 were measured in serum samples, as well as complement factor protein C4a and FGF21, which are exported into the circulation by the liver in response to stress. There were no notable statistical variances detected in plasma C4a levels when comparing individuals with ME/CFS to healthy participants (Figure 2A). However, upon stratifying patients based on their immunological status, a noteworthy, statistically significant decrease in plasma C4a concentrations was observed in individuals with ME/CFS and immunodeficiencies (ME/CFS + ID) in contrast to those with ME/CFS but no immunodeficiencies (ME/CFS − ID) and the healthy control group (Figure 2B). A significant reduction in serum IL-1-beta levels was observed when comparing ME/CFS patients without immunodeficiencies to the healthy control group (Figure 3A). Serum IL-33 levels were increased in ME/CFS patients without immunodeficiencies when compared to healthy control participants but also slightly increased when compared to ME/CFS patients with immunodeficiencies (Table A2). A similar pattern was observed in serum levels of IL-6 as shown in Figure 3B, with a statistically significant difference in the ME/CFS − ID group compared to healthy controls. Evaluation of TNF-alpha, IL-8, and FGF21 in serum did not show statistically significant differences between the tested groups as described in Table A2. The data on the IL-8 and IFN-gamma levels are not shown, as the values neither show statistical significance nor a disease-specific pattern.

### 3.2. Immune Marker Related to Enhanced Mast Cell Activity and Eosinophil Activation

As chronic mast cell activation is difficult to determine, due to the acute nature and short half-life of the major mediators tryptase and histamine [24], we aimed to measure the serum levels of ECP and EDN. The measurement of ECP did not reveal differences between the test groups of the study cohort and thus is not shown. Serum EDN levels were more elevated in ME/CFS patients with immunodeficiencies (50.11 ± 27.60 ng/mL) and without immunodeficiencies (52.29 ± 29.19 ng/mL) than in the healthy control group (41.03 ± 17.78 ng/mL, Table A2). The results are not statistically significant.

### 3.3. Biomarker Associated with Mucosal and Intestinal Barrier Integrity

When comparing ME/CFS patients to the healthy control participants (Figure 4A), there were no statistically significant differences detected in serum Lipopolysaccharide-binding protein (LBP) levels. However, following immunological patient stratification, serum LBP levels were found to be significantly increased in patients without immunodeficiencies (ME/CFS − ID) when compared to the healthy control group (Figure 4B). The soluble CD14 serum level, which is needed as a co-factor together with LBP to mediate innate immunity against LPS in the immune system, did not differ between the test groups (Table A2). However, when comparing the LBP/sCD14 ratio, we were again able to only detect a significantly higher ratio in ME/CFS patients without immunodeficiencies compared to the other test groups (Figure 4C,D). I-FABP levels were slightly reduced in ME/CFS patients without immunodeficiencies compared to healthy control participants, but the results were not significant (Table A2). Endotoxin-core IgG-antibodies were highest in immunodeficient ME/CFS patients (113.10 ± 219.30 MU/mL), while ME/CFS patients without immunodeficiencies had similar Endotoxin-core IgG levels as healthy control participants (Table A2), although the results are not statistically significant.

## 4. Discussion

Previous studies on ME/CFS already investigated immunological dysregulation with the potential to trigger inflammatory processes [2,11]. In our research, we focused on investigating the mucosal intestinal barrier to gain a deeper understanding of pathological mechanisms underlying ME/CFS. Our goal was to bridge these two aspects to elucidate the link between mucosal (intestinal) barrier function and immunodeficiencies. We divided the ME/CFS patients into two subgroups based on immunological stratification. Based on our results, we defined three key messages. Our data suggest that the two patient groups exhibit distinct pathophysiological mechanisms, and most importantly, we show the importance of immunological stratification prior to down-stream analyses. Our findings should be considered when diagnosing ME/CFS and developing targeted medications to mitigate disease progression and improve patient health care.

It was previously demonstrated that C4a levels in ME/CFS patients were in general lower than in healthy control participants but elevated after exercise challenge and correlated with PEM-associated symptoms [25,26,27]. Interestingly, the rise in C4a levels was not observable one hour after being physically challenged [25,26,27] but only after six hours [27] and returned almost to baseline 24 h post-exercise. This suggests C4a as a potential PEM-associated biomarker and a pathophysiological marker in ME/CFS. It is noteworthy that none of the above-mentioned studies examined the immune competence of the ME/CFS patients before study participation. In our study, comparing all ME/CFS patients to healthy control participants showed reduced C4a levels as well, although the results were not significant. However, after stratifying ME/CFS patients based on their immune competence, we found statistically reduced C4a levels in ME/CFS patients with immunodeficiencies compared to both other test groups (Figure 2). Furthermore, ME/CFS patients without immunodeficiencies showed C4a levels comparable to healthy controls. This demonstrates that not only is immunological patient stratification required before study inclusion when using C4a as marker of interest, but it also indicates a potential dysregulation in the inflammatory processes involving complement system activation. Ten percent of the ME/CFS patients with immunodeficiencies in our study suffered from a complement deficiency (Figure 1). C4a, which serves together with C3a and C5a as an inflammatory anaphylatoxin, plays a crucial role in histamine release from mast cells, increased vascular permeability, chemotaxis, inflammation, smooth muscle cell contraction, and the generation of cytotoxic oxygen radicals [28]. These effects may contribute to the flu-like symptoms frequently associated with PEM.

Elevated levels of the barrier marker LBP indicate mucosal barrier leakage in the ME/CFS patient group without immunodeficiencies (Figure 4B). This marker is associated with bacterial wall components that breach the intestinal barrier and enter the bloodstream [29]. LBP production is dependent on circulating LPS concentrations in the blood, as the body produces more LBP with increased LPS levels. Interestingly, we did not find sCD14 levels to be different within the whole study cohort (Table A2), although it is needed as a co-factor for LBP when mediating innate immunity [29]. In mice, it has been demonstrated that sCD14 presents protective effects in inflammatory bowel disease and that a rise in the LBP/sCD14 ratio in both humans and mice is linked to an increase in plasma IL-6 [30]. Thus, it is suggested that LBP and sCD14, as well as the ratio to each other, are important factors in low-grade inflammation during metabolic diseases. In line with these data, the ME/CFS − ID group, which has the highest LBP values, also shows a higher LBP/sCD14 ratio. Furthermore, we could see a higher level of serum IL-6 (Figure 3B) compared to the other study groups, but the individual samples do not correlate with each other when comparing the patterns of the LBP and IL-6 measurements. Our data indicate a barrier leakage, but to which degree this contributes to low-grade inflammation has to be further studied. Also, our findings regarding IL-1-beta (Figure 3A) and IL-33 levels (Table A2), which are associated with homeostasis of the gut microbiota, support the hypothesis of a mucosal barrier leakage [31,32]. Gut microbiota homeostasis is essential for the maintenance and repair of the mucosal epithelial barrier. IL-1-beta production induced by mucosa-associated gut commensal bacteria promotes intestinal barrier repair, thus reduced IL-1beta levels in ME/CFS might promote barrier dysfunction. Increased IL-33, which is a member of the IL-1 cytokine family, points towards a mucosal barrier leakage in ME/CFS patients without immunodeficiencies, as it acts as an alarmin to provide acute initial protection [31]. However, the IL-33 results were not statistically significant and studies with larger patient cohorts would be needed to verify this observation. It is important to emphasize that intestinal barrier leakage is not necessarily induced by epithelial damage, as the gastrointestinal mucosal tissues also regulate its dynamics via intra- and inter-cellular transportation pathways [33] The endotoxin-core antibodies and I-FABP level, which are found in elevated levels in situations of epithelial damage [34], do not statistically differ between the test groups. Altogether, we hypothesize that ME/CFS patients without immunodeficiencies suffer rather from epithelial leakage, as indicated by the elevated LBP level and the LBP/sCD14 ratio, and ultimately fail to maintain mucosal barrier function.

Previous research on ME/CFS has often focused on altered cytokine patterns, mucosal barrier dysfunction, chronic low-grade inflammation, and various other mechanisms, but the results have been inconsistent. The multi-factorial nature and the still unknown disease trigger hamper research as much as the highly individual symptoms, which also differ depending on the disease severity and co-morbidities of affected patients. To support this observation, we aimed to bridge immunological and mucosal barrier dysfunction as the gut-associated lymphoid tissues represent the largest immune organ in the human body [35]. No major differences were found in most measured parameters when comparing ME/CFS patients with healthy control participants. However, an interesting result was detected when examining the immunological marker C4a (Figure 2A) and the mucosal barrier marker LBP (Figure 4A,C). If ME/CFS patients are not stratified based on their immune competence, the measured results are averaged out and comparable to the healthy control participants (Figure 2A and Figure 4A,C). Thus, we provide evidence that ME/CFS patient subgroups exhibit different pathophysiological patterns. These distinct patterns represent subgroup-specific characteristics, and thus they might serve as target points for future diagnostic and treatment approaches. Without precise patient stratification, potential biomarkers might lose discrimination power.

As patient recovery is poorly and incoherently documented [36], it is difficult to find common mechanistic pathways. Attempts to define causative treatment approaches have failed so far. From an immunological perspective, promising treatment strategies could be identified in well-characterized ME/CFS subgroups [37]. When it comes to the gastrointestinal barrier function, treatment approaches are rather based on case reports and off-label medication, partially due to highly individual lifestyle factors. More research needs to be conducted to successfully translate gastrointestinal barrier strengthening factors, like immune–nutritional approaches, into applied clinics. Altogether, these findings highlight the importance of well-stratified ME/CFS patients in research and medical care. Promising treatment approaches can be defined on subgroup-specific characteristics but might fail if patient subgroups are too heterogenous [37].

## 5. Conclusions

This research investigates the relationship between immunological abnormalities and mucosal intestinal barrier function in ME/CFS patients. The study stratified ME/CFS patients into two groups based on their immunological status and presented three key findings: Reduced C4a levels in ME/CFS patients with immunodeficiencies suggested a subgroup-specific disease pattern impairing innate immunity. It remains elusive if this deficiency contributes to disease onset due to a failure of innate immunity in situations of (viral) host defense or is a result of the ongoing disease. ME/CFS patients without immunodeficiency exhibited mucosal barrier leakage, indicated by elevated levels of the marker LBP, potentially contributing to low-grade inflammation.

The use of standardized detection methods is needed to rapidly translate research into clinics. This ensures the definition of bench-to-bedside orientated approaches to make evidence-based ME/CFS diagnosis and treatment generally available for patients. Stratifying ME/CFS patients based on their immune competence enabled the distinction of ME/CFS patient subgroups with generally accessible detection methods. This finding further highlights the importance of careful patient stratification when studying ME/CFS, especially in the evaluation of novel treatment strategies, as different disease mechanisms might be involved in fatigue and PEM development. However, the tested cohorts were relatively small, which presents a major limitation of this study. Larger study cohorts are needed to verify or decline the results, which are not significant but might indicate a trend. We further acknowledge that our study would benefit from a more detailed description of patients’ illness characteristics. The immunological evaluation highlighted the importance of a detailed stratification of this highly heterogenous patient group. Especially when examining small study cohorts, a thorough description is crucial for data interpretation and reproducibility.

The COVID-19 pandemic clearly demonstrated that viral infections may be of asymptomatic nature [38], highlighting the underestimated danger arising from post-infectious diseases. Due to the progression of climate change, novel pathogens, like the West Nile virus or tick-borne diseases, which trigger post-infectious conditions, are on the forefront [4,39]. The population of ME/CFS patients will increase and will include more patients who experience an asymptomatic disease course and are not aware of their infection. The role of immune dysregulation and mucosal barrier disruption as disease mechanisms needs in-depth investigations as they represent promising target point in ME/CFS and post-viral treatment. When considering that around 25% of ME/CFS patients are bedbound and 60% of patients are unable to work full-time, it is obvious that prolonged fatigue represents an enormous health burden and socioeconomic challenge [40]. ME/CFS is associated with yearly costs of 17–24 billion USD in the United States alone [41], highlighting the enormous impact of the disease. It is noteworthy that this information is based on an estimation from 2004. Considering the drastically increasing incidences of post-virally triggered ME/CFS due to SARS-CoV-2 infections, the financial burden for health authorities, as well as individual patients, needs to be counteracted as soon as possible.

## Figures and Tables

**Figure 1 jcm-13-00275-f001:**
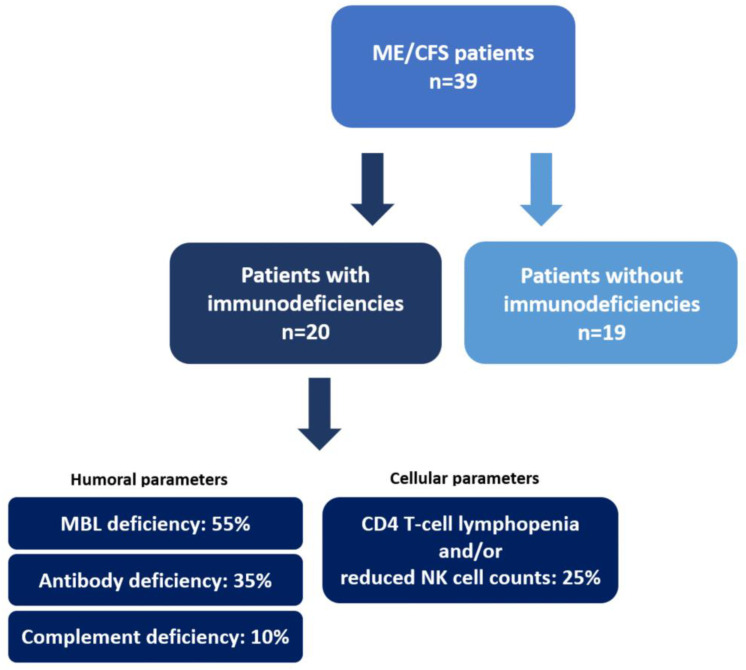
Patient stratification based on immunological parameters. The information on immunodeficiencies of patients was obtained during medical examinations of already diagnosed ME/CFS patients prior to study participation. All patients were recruited in the doctor’s office of Assoc.-Prof. Eva Untersmayr.

**Figure 2 jcm-13-00275-f002:**
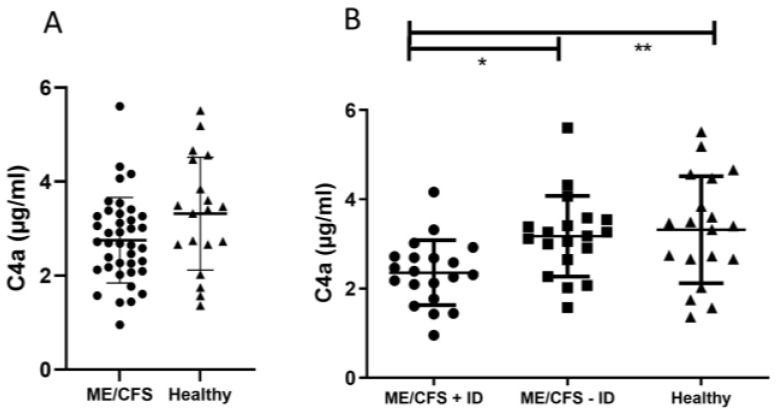
(**A**) No statistically significant differences in plasma C4a level were found comparing ME/CFS patients with healthy controls. (**B**) After immunological patient stratification, concentrations of plasma C4a samples are significantly reduced in patients with immunodeficiencies (ME/CFS + ID) compared to patients without immunodeficiencies (ME/CFS − ID) and healthy control participants. LOD: 6 pg/mL; * *p* < 0.05. ** *p* < 0.01.

**Figure 3 jcm-13-00275-f003:**
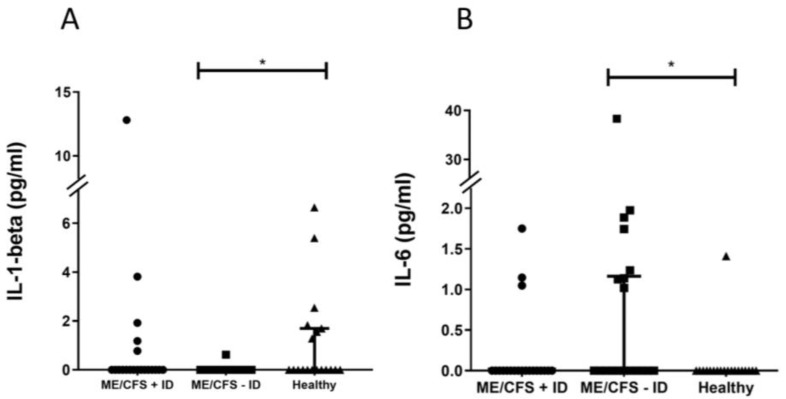
(**A**) Significantly reduced serum IL-1-beta levels were found when comparing ME/CFS patients without immunodeficiencies to healthy control participants. (**B**) Serum IL-6 levels were significantly elevated in ME/CFS patients without immunodeficiencies compared to healthy control participants. LOD: 2 pg/mL; * *p* < 0.05.

**Figure 4 jcm-13-00275-f004:**
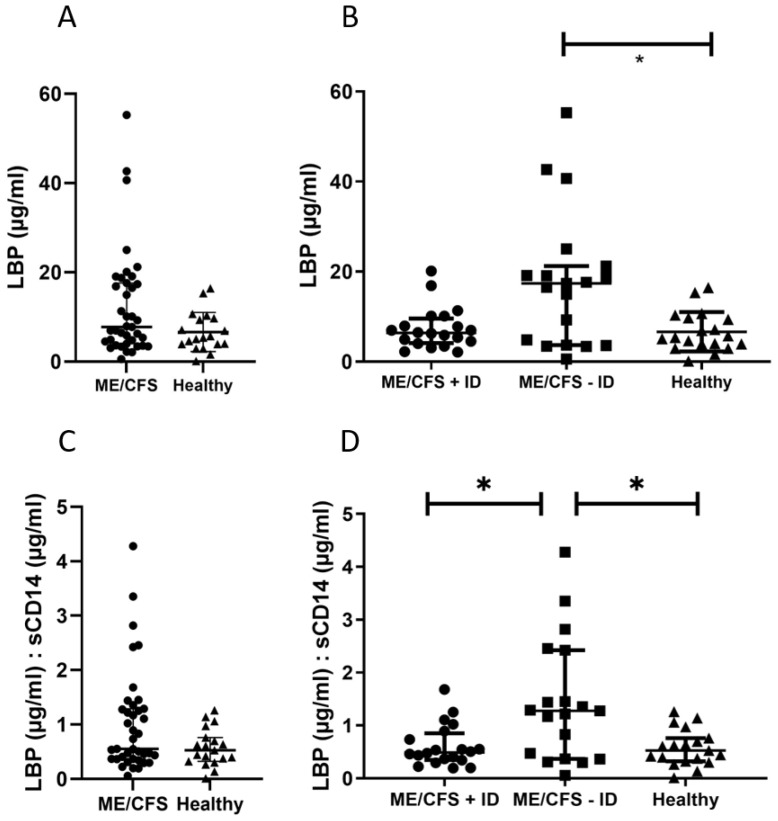
(**A**) When comparing ME/CFS patients to healthy control participants, no statistically significant differences in serum LBP level were found. (**B**) Only after immunological patient stratification was serum LBP level significantly elevated in patients without immunodeficiencies (ME/CFS − ID) compared to healthy control participants. (**C**) A similar effect was detected when comparing the LBP/sCD14 ratio of all ME/CFS patients to healthy control participants. (**D**) Again, only after immunological stratification did ME/CFS patients without immunodeficiencies show a significantly higher LBP/sCD14 ratio compared to the other test groups. LOD (LBP): 0.82 ng/mL; * *p* < 0.05.

**Table 1 jcm-13-00275-t001:** Demographics and in-/exclusion criteria of the study cohort. * ME/CFS was diagnosed based on the IOM criteria [22] and includes a profound and debilitating fatigue, PEM, unrefreshing sleep, and at least one of the two following manifestations: cognitive impairment and/or orthostatic intolerance.

Study Group	ME/CFS + ID	ME/CFS − ID	Healthy
**Demographic Data**			
Mean age in years (±SD)	41.2 (±12.6)	38.4 (±10.8)	43.1 (±13.0)
Female sex in percentage (n)	75 (15)	84.2 (16)	73.7 (14)
**Inclusion criteria**			
Immunodeficiency (ID)	yes	no	no
ME/CFS (IOM criteria) *	yes	yes	no
**Exclusion criteria**			
Neurological/psychiatric co-morbidities	no	no	yes

**Table 2 jcm-13-00275-t002:** Co-morbidities, which have been diagnosed before study participation and do not exclude an ME/CFS diagnosis, are listed. Results are presented in percentages.

Study Group	ME/CFS + ID	ME/CFS − ID	Healthy
**Co-morbidities (%)**			
Fibromyalgia	5.0	0.0	0.0
Postural orthostatic tachycardia syndrome	20.0	52.6	0.0
Orthostatic dysregulation	45.0	78.9	5.3
Irritable bowel syndrome	30.0	21.1	0.0
Food intolerances/atopic conditions	70.0	68.4	42.1
Mild anxiety	0.0	10.5	0.0
Mild depression	15.0	21.1	0.0
Hypermobility Ehlers–Danlos syndrome	0.0	10.5	0.0
Small fiber neuropathy	15.0	31.6	0.0
Endometriosis (female cohort)	26.7	6.25	0.0
Chronic pelvic pain	5.0	0.0	5.3
Irritable bladder	5.0	21.1	0.0
Hashimoto thyroiditis/hypothyroidism *	20.0	0.0	5.3
Mast cell activation syndrome	20.0	36.8	0.0

* Clinically controlled.

## Data Availability

Further information is available from the corresponding authors upon reasonable request.

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
