# Peer review of "Immunological Patient Stratification in Myalgic Encephalomyelitis/Chronic Fatigue Syndrome"

_jcm, 2024, doi:10.3390/jcm13010275_

Round 1

Reviewer 1 Report

Comments and Suggestions for Authors

The authors present data on a small cohort from one clinical practice of patients with ME/CFS (n=39) and healthy controls (n=19) suggesting that subgrouping ME/CFS based on immunodeficiencies identifies two ME/CFS groups with different pathophysiologic patterns. The observation is very interesting and emphasizes the importance of studies subgrouping patients based on biologic markers in order to better define underlying pathology and treatment options. The authors are encouraged to address several questions to improve the clarity and validity of their message.

1. 1.The introduction is very interesting but very broad. In order to allow more space to fully describe their study and findings, suggest omitting most of paragraphs 2-4.

2.       Additional information about the study cohort is needed:

a.       When were participants enrolled? Was there any potential for SARS CoV-2 impact? What proportion of eligible patients were enrolled?

b.       Was the study protocol reviewed and approved by Human Subjects Institutional Review Board (or similar process for ethical approval)? Was informed consent obtained?

c.       How was the IOM clinical case definition applied? It appears to be based on clinical opinion, which is fine, just should be stated. As patients were recruited from one clinical practice they were presumably evaluated for other medical and psychological conditions that could contribute to symptoms, but it would be helpful to state this (and perhaps clinic’s approach to screening, differential diagnosis)

d.       Identification of the immunodeficient subgroup is a key part of the paper, but the specific measures are only included in the figures. Each measure needs to be defined and method for detection included. How were these specific measures selected and what threshold was used to indicate a deficiency? Are population norms established for these values – if so, how do thresholds compare for these?

e.       What is the justification for combining four apparently disparate measures of immune function into a group of ‘immunodeficient’? How much difference in these measures is seen in the group without immunodeficiencies?

f.        Additional information about illness characteristics of the group overall and divided by immune status is needed. Basic items such as duration of illness, mode of onset, some functional measures, measures of fatigue and other symptoms, and comorbidities.

3.       Additional information about assay performance is required. Were assays validated in the laboratory prior to use (helpful in supplement)? What was the coefficient of variation between different runs of the assay and between lots of kits? Were samples tested in replicates (if so how many) and dilution series? Was there a requirement that dilutions provided linear results? Were all samples tested in same run? If multiple runs, over what time frame?

4.       Data in Figure 3 are hard to interpret due to one outlier high value. For IL-1beta in the ME/CFS + ID group and for IL-6 in the ME/CFS-ID group. In both cases this one sample appears to account for the statistical significance. This should be evaluated and data shown on scale that permits any spread in the lower values to be observed (perhaps by using interrupted scale for the outlier point).

5.       Discussion includes information from other studies about the importance of LBP/sCD14 ratio. As the data are available in this study, they should provide this result.

6.       Discussion includes statements relating mean values of analytes to each other, for example ME/CFS-ID has highest LBP values, also shows higher level of IL-6. If such comparisons are made, they should be based on well values from individual samples correlate.

7.       Discussion should focus on findings in this report, while other paragraphs would benefit from focus as well, the last paragraph of discussion seems out of scope. Discussion points are included in the conclusion section.

8.       Discussing group differences in analytes that are presented in results as not significantly different is not justified. If there are differences that don’t reach statistical significance but may be worth keeping in mind as following expected trends, these need to be included in results.

9. Study strengths and limitations should be included in discussion.

Author Response

We would like to thank Reviewer 1 for the thorough evaluation of our manuscript. We appreciate the time and efforts, which you made. Please find the corrections based on your comments in the attached point-by-point reply. To avoid any miscommunication, we would like to point out that the line numbers are referred to the revised manuscript with the marked changes.

Reviewer 2 Report

Comments and Suggestions for Authors

The paper needs to be improved as follows:

1. Introduction: Not all studies of ME/CFS show dysregulated NK cells. and abnormalities of cytokines.  Some cases of ME/CFS do have known triggers (eg infectious mono, Q fever, Ross River Virus, etc.) 

2. Materials and Methods: Authors never state what constitutes immunodeficiency in their patients with ME/CFS.  Authors do not state how they classified patients with ME/CFS as mild-moderate vs severe.  It doesn't seem that healthy controls were excluded from neurological or psychiatric pre-conditions from Table 1.  P values and statistics should be given in Table 1.  

3. Results: Some of this material should be in the Methods section (eg lines 203-207).  Authors need to explain why sometimes differences were seen in the Me/CFS+ID group and sometimes in the ME/CFS-ID group; it is possible that by just dividing the group in 1/2 the authors have increased the chances of finding something due to multiple comparisons.  It is also unclear why some results are presented in Tables and some in Figures.  No rationales are given for the analysis of ECP, EDN, LBP, L-FABP and endotoxin-core-IgG-antibodies.

4. Discussion: No data are shown for the statement in lines 296-297 and lines 330-331.  Why would barrier leakage and low grade inflammation occur in patients with ME/CFS without immunodeficiency (Lines 314-319)?  MIS-C is different from ME/CFS and thus the discussion in lines 369-373 should be deleted.  

5. Conclusion: There was no effort to distinguish fatigue and PEM in the current paper (lines 392-394).   MANY viruses can be asymptomatic (Lines 396-397).  Line 403-404 needs a reference.  

6. Reference 21 is not complete.

Comments on the Quality of English Language

Could use some English editing.

Author Response

We would like to thank Reviewer 2 for the thorough evaluation of our manuscript. We appreciate the time and efforts, which you made. Please find the corrections based on your comments in the attached point-by-point reply. To avoid any miscommunication, we would like to point out that the line numbers are referred to the revised manuscript with the marked changes.

Round 2

Reviewer 1 Report

Comments and Suggestions for Authors

The authors have responded to the comments, but a few need additional attention.

T    1. The additional explanation of the tests used for identification of immunodeficiency is helpful, but the threshold for each measure remains ambiguous. The methods state that the group without immunodeficiencies were within the normal range of these tests. This implies that that the immunodeficiency group cut-off was simply the lower bound of the normal range. Given that the normal range is generally set at 95% CI of populations measures, detecting a clinically significant abnormality may require use of a lower value. It would be helpful to explain clearly the threshold used and rationale for the choice.

2   2. The additional information about the illness characteristics of the groups divided by immune status is provided in Table 2. However, information on duration of ME/CFS, type of onset, and functional status is missing. If this is available it would be helpful to include. Collection of illness characteristics using ME/CFS Common Data elements should be considered in future research. https://www.commondataelements.ninds.nih.gov/Myalgic%20Encephalomyelitis/Chronic%20Fatigue%20Syndrome Also, Table 2 is apparently percentages, but this is not specified. Were any of these factors statistically different between groups?

3   3. While the definition of severity of illness is referenced, it would be helpful to describe the definition or provide table or description in supplemental material.

4   4. The additional information about the assays is helpful. Able to reproduce results in important. What was the coefficient of variation in measures of samples conducted on two different days? Given the time-frame of the study, what is known about stability of these analytes during storage? Was time to assay reviewed as a variable in results? Was each assay verified to provide linear results in diluted samples?

5   5. Limitations of the study should be expanded, depending on availability of information requested on additional illness characteristics, assay replication, and thresholds for clinically meaningful immunodeficiency. In addition to small cohort, enrolling from one clinic can limit generalizability in absence of standardized approach to case identification and management.

Author Response

We would like to thank again reviewer 1 for the fast and comprehensive review of our manuscript. We appreciate the invested time and efforts to improve the quality of our study.

Reviewer 2 Report

Comments and Suggestions for Authors

Paper is improved. 

However, there are still many concerns:

1. There is no clear rationale for the testing done.  For example, why did the authors not choose to measure other markers of mucosal barrier integrity such as Il17 or IL 22?  And why did they examine EDN, ECP LBP and CD14?  I agree that I-FABP is relevant to intestinal barrier dysfunction.

2. There is no statement in the Methods section of all of the reagents used.  In fact, I-FABP is not mentioned till the end of the Results section and is never spelled out anywhere in the paper that I could see. 

3. The immune deficiencies the authors discuss are quite common in the general population.  Were the healthy controls screened for NOT having any of them?

4. I am not sure what "leukocyte composition" means in Fig. 1.

5. Which "complement factors" were deficient in the patients with ME/CFS and immune deficiencies (Fig. 1)?  If it was C4a, then of course one would expect the results shown in Fig. 2B.   

6. Many results are discussed that are not statistically significant.

7. Even if it is true that only patients with ME/CFS and no immune deficiencies have intestinal barrier pathology, a rationale as to why that might be should be provided.

Comments on the Quality of English Language

English is OK.

Author Response

We would like to thank again reviewer 2 for the fast and comprehensive review of our manuscript. We appreciate the invested time and efforts to improve the quality of our study.
